# Factors associating different antenatal care contacts of women: A cross-sectional analysis of Bangladesh demographic and health survey 2014 data

Sanjoy Kumar Chanda[1,2]*, Benojir Ahammed[3], Md. Hasan Howlader[4], Md Ashikuzzaman[4], Taufiq-E-Ahmed Shovo[2,5], Md. Tanvir Hossain[2]

1 School of Healthcare, Faculty of Medicine and Health, University of Leeds, Leeds, England, United Kingdom, 2 Sociology Discipline, Social Science School, Khulna University, Khulna, Bangladesh, 3 Statistics Discipline, Science, Engineering and Technology School, Khulna University, Khulna, Bangladesh, 4 Development Studies Discipline, Social Science School, Khulna University, Khulna, Bangladesh, 5 School of Humanities and Social Science, Faculty of Education and Arts, University of Newcastle, Callaghan, New South Wales, Australia

* hcskch@leeds.ac.uk

**Data Availability Statement:** The data underlying the results presented in the study are available from BDHS-2014. https://dhsprogram.com/

## Abstract

Antenatal care (ANC) contacts have long been considered a critical component of the continuum of care for a pregnant mother along with the newborn baby. The latest maternal mortality survey in Bangladesh suggests that progress in reducing maternal mortality has stalled as only 37% of pregnant women have attended at least four ANC contacts. This paper aims to determine what factors are associated with ANC contacts for women in Bangladesh. We analysed the data, provided by Bangladesh demographic and health survey 2014, covering a nationally representative sample of 17,863 ever married women aged 15–49 years. A two-stage stratified cluster sampling was used to collect the data. Data derived from 4,475 mothers who gave birth in the three years preceding the survey. Descriptive, inferential, and multivariate statistical techniques were used to analyse the data. An overall 78.4% of women had ANC contacts, but the WHO recommended ≥8 ANC contacts and ANC contacts by qualified doctors were only 8% for each. The logistic regression analysis revealed that division, maternal age, women's education, husband's education, wealth index and media exposure were associated with the ANC contacts. Likewise, place of residence, women's education, religion, and wealth index were also found to be associated with the WHO recommended ANC contacts. Furthermore, the husband's education, division, religion and husband's employment showed significant associations with ANC contacts by qualified doctors. However, Bangladeshi women in general revealed an unsatisfactory level of ANC contacts, the WHO recommended as well as ANC contacts by qualified doctors. In order to improve the situation, it is necessary to follow the most recent ANC contacts recommended by the WHO and to contact the qualified doctors. Moreover, an improvement in education as well as access to information along with an increase of transports, care centres and reduction of service costs would see an improvement of ANC contacts in Bangladesh.

publications/publication-fr311-dhs-final-reports.
cfm.

**Funding:** We did not receive any funding for this
work.

**Competing interests:** The authors have declared
that no competing interests exist.

## Introduction

The United Nations (UN) Sustainable Development Goals (SDGs) call for a global reduction
of maternal mortality to 70 or less per 100,000 live births and ensuring universal access to sex-
ual and reproductive healthcare services by 2030 [1]. Based on a systematic analysis by the UN
Maternal Mortality Estimation Inter-Agency Group, in 2015 alone, approximately 830 women
die every day globally due to complications during pregnancy or childbirth; around 99% of
these deaths take place in developing countries [2, 3, 18]. A range of studies indicates that
access to quality antenatal care can avoid maternal deaths [4, 5] and it can reduce up to 20%
maternal mortality [6, 17]. ANC services have long been considered a critical component of
the continuum of care for pregnant mothers along with newborns [7]. ANC has been consid-
ered an important service in ensuring safe motherhood as the world goes through an obstetric
transition [8, 10].

ANC has been defined as care given by skilled individuals with extensive healthcare training
to both pregnant women and adolescent girls to ensure the best possible health conditions for
both the mother and foetus during gestation [5]. ANC provides a platform for essential health-
care functions, including health promotion, screening and diagnosis, and disease prevention
[5]. The purpose of ANC is to monitor and safeguard the wellbeing of the mother and foetus,
detect any pregnancy complications and take necessary measures, respond to mother's com-
plaints, prepare a mother for birth, and promote healthy behaviours of mothers [9]. Globally,
it is observed a continuous growth of ANC utilisation throughout the past decades, and now a
significant portion of women (86%) is attending at least one ANC contact and from concep-
tion to birth, 62% receiving at least four ANC contacts [10]. However, over the last two
decades, the ANC utilisation has increased remarkably but the quality of such ANC services
has remained poor to some extent that demand rigorous scrutiny as these poor quality services
compromise the potential benefits of getting such cares. [11]. Thus, the World Health Organi-
zation (WHO) [5] recently recommends eight ANC contacts instead of earlier four contacts to
ensure positive pregnancy for expected mothers.

Bangladesh is a developing country, and the health status of this country is now better com-
pared to the past decades. This country has achieved a notable advancement in achieving the
Millennium Development Goals, contributing to the reduction of maternal deaths, and at the
moment working to the newly agreed SDGs to be fulfilled by 2030 [12]. However, the full
potential of maternal health services has never been met [10, 13]. The latest Bangladesh mater-
nal mortality survey suggests that progress in reducing maternal mortality has stalled, and only
37% of pregnant women attend at least four ANC contacts [14]. Although the government of
Bangladesh along with nongovernmental and international organisations are working together
to increase the number of ANC contacts, the achievement is not remarkable [15].

Earlier studies conducted in Bangladesh and South Asia have mostly categorized relevant
factors associated with ANC contacts of pregnant women as demographic, socioeconomic
and environmental [15–23]. Only a handful of studies, using data from previous demo-
graphic and health surveys, were conducted to find out the association between determi-
nants and contents of ANC contacts in Bangladesh [10, 17]. Although there is a range of
works on ANC services, the main focus was on disclosing factors associated with ANC con-
tacts and the use of ANC contents. However, few crucial issues, such as the number of ANC
contacts, [17] the new WHO recommended more than or equal to 8 times ANC contacts
[17] and ANC contacts by professionals, were minimally addressed by the health researchers.
The aim of this paper is therefore to explore the factors associated with the ANC of women
in Bangladesh.

## Materials and methods

### Data source

We used data from the 2014 Bangladesh demographic and health survey (BDHS) [24], which is the national level population and health surveys conducted as part of the global demographic and health survey (DHS) programme. The 2014 BDHS was conducted by the National Institute of Population Research and Training (NIPORT) of Bangladesh, Mitra and Associates, a Bangladesh-based research firm. ICF International of Calverton, Maryland, USA, provided technical assistance for the survey as a part of its international demographic and health survey programme (MEASURE DHS) and the U.S. Agency for International Development (USAID) provided financial support to complete the survey. The BDHS-2014 database is available online and can be retrieved through registration. We retrieved and utilised this data with permission from the DHS programme. The BDHS-2014 was a retrospective survey based on a two-stage stratified cluster sampling design, where each of the seven administrative divisions was treated as strata. The survey covered a nationally representative sample of 17,863 ever married women aged 15–49 years, who were interviewed from 17,300 randomly selected households. We considered all these 17,863 ever married women for the current study. Information was collected about the person and institution providing ANC, the number of ANC contacts and the items included in the ANC delivered. The sample therefore for this study was 4475 women.

### Analytical framework

We adopted the modified framework of Andersen's behavioural model [25] for ANC utilisation, which was previously used to underpin health research in Bangladesh [15, 26]. In the model, a wide range of variables affecting ANC services are outlined in three factors: the geographical environment, predisposing and enabling factors. The geographical environment factors cover the state of the physical environment, such as division and participant's place of residence. The predisposing factors reflect the propensity of individuals, and it includes characteristics, particularly age, birth order and education. Finally, enabling factors represent the actual ability of the individual to obtain healthcare services that include media exposure, wealth index and employment. It is argued that the predisposing factors reflect the fact that families with different characteristics have different propensities to use healthcare services. In contrast, the enabling factors reflect the fact that some families, even if predisposed to use health services, must have some means to obtain them, i.e., income, access and availability of health services [27]. The three factors of the model are outlined in Fig 1, together with the associated variables used in the current study. In this paper, we used the word 'contact' instead of 'visit' as recently the WHO used the word 'contact' as it implies an active connection between a pregnant woman and a healthcare provider that is not implicit with the word 'contact' [5].

### Variables

This study included two types of variables, such as dependent and independent. Dependent variables of this study were threefold: (1) the number of ANC contacts [17], (2) the WHO recommended ≥8 ANC contacts [17] and (3) ANC contacts by qualified doctors. The WHOs recently arrived at a new model, the 2016 WHO ANC model, where it focused on the eight contacts to ensure positive pregnancy–that replaces the four-contact focused ANC model [5].

Among independent variables, we included divisions and place of residence (rural/urban) as external geographical factors. Several studies identified the place of residence as a significant determinant for healthcare access and utilisation of married women [15, 28]. The 2014 BDHS

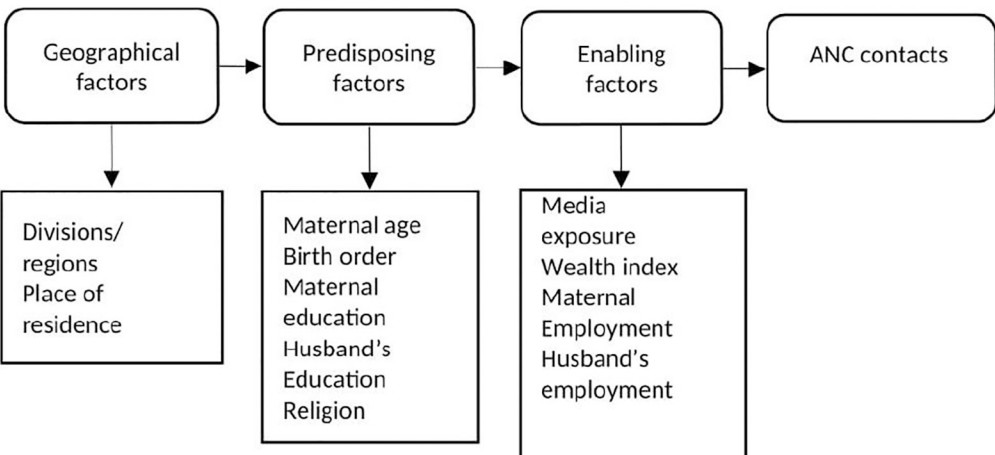

**Fig 1. Analytical framework of factors associated with ANC contacts in Bangladesh.** Adapted from Andersen [25].

included data of seven administrative divisions of Bangladesh: Barisal, Chittagong, Dhaka, Khulna, Rajshahi, Rangpur and Sylhet. All these regions are geographically unique, for instance, flood-prone areas (Barisal and Dhaka), hilly regions (Chittagong and Sylhet), cyclone-prone areas (Khulna and Barisal) and *manga* (seasonal food scarcity) areas (Rajshahi and Rangpur) [15]. These variations may likely have an impact on healthcare needs and utilisation [26]. We included maternal age, birth order, education and religion as predisposing factors. Maternal age [18] and birth order [15] have been found as significant predisposing determinants for maternal healthcare services in Bangladesh. Education was categorized into four major categories, including no education, primary, secondary and higher.

Earlier studies have identified the importance of maternal education for the utilisation of healthcare services in Bangladesh [17, 23]. In a study by Simkhada et al. [29], they have observed that women among the Muslims are more likely to contact ANC services compared to other religious sects in developing countries. Thus, we identified religion as a predisposing factor, and this variable was divided into Islam and others.

Enabling factors included media exposure, which was categorized into threefold: not at all, less than once a week and at least once a week. Wealth index included five categories, e.g. poorest, poorer, middle, richer and richest. The employment variable of both the mother and her husband was dichotomized into working and non-working categories.

## Data analysis

Data were analysed using three levels of statistical analysis: univariate, bivariate and multivariate. Characteristics of women were identified using percentage distribution. Bivariate analysis, as simple summary statistics, was employed to determine the statistically significant relationship at $p<0.05$ between dependent variables and selected explanatory variables. The statistical significance was tested by Pearson's chi-square ($\chi^2$) test of independence for categorical dependent variables. Finally, the binary logistic regression models were executed as the dependent variables were categorical. These models were performed with the variables which were found statistically significant at 5% in bivariate analyses. The results of the binary logistic regression analyses were shown using odds ratios (OR) with 95% confidence intervals (CI). All the statistical analyses were performed in SPSS v25.

### Ethical issues

We used nationally representative data, widely used for public health issues, extracted from the BDHS-2014. Thus, it was not necessary to get ethical approval from any institution in Bangladesh.

## Results

### Background characteristics of participants

It is evident that around one-fifth of the respondents (19.2%) lived in Chittagong, followed by Dhaka (17.6%), Sylhet (15.1%), Rangpur (12.3%), Rajshahi (12.1%), Barisal (11.8%) and Khulna (11.8%) (Table 1). About 70% of the respondents resided in rural areas and nearly three quarters of them belonged to the age group of 20–34 years. The majority (70.5%) reportedly have less than three children. Among the respondents, mostly from Muslim families (92%), about half of them (47.4%) have no formal education, and only one-fifth (21.9%) were involved in income-generating activities. Only a quarter percent of the spouses had secondary or higher education; however, nearly cent percent of them (97%) were involved in economically productive activities. Around half of the respondents (49.9%) had access to media.

### Association of factors using bivariate analysis

Table 1 also presents the bivariate analysis of respondents' ANC contacts, the WHO recommended ≥8 ANC contacts and ANC contacts by qualified doctors. Overall, 78.4% of women contacted for ANC services, but only 8.0% of women fulfilled the WHO recommended number of ANC contacts and 8.1% women did not receive ANC services by qualified doctors.

The maternal education, husband's education and wealth index were significantly associated with ANC contacts, the WHO recommended ANC contacts and ANC contacts by qualified doctors. All types of contact for ANC services, however, were higher among the highly educated women, women from richest families and women having educated spouses.

The ANC contacts and WHO recommended ≥8 ANC contacts among women in Khulna division (88.5% and 10.0%) were significantly higher than other divisions. Still, the ANC contacts by qualified doctors among women were relatively higher in Sylhet division (11.9%). The prevalence of ANC contacts and WHO recommended ANC contacts were significantly higher among urban women than their rural counterparts. Maternal age, birth order and media exposure were also significantly associated with ANC contacts and the WHO recommended ANC contacts. Compared to other age groups, women from 35–49 age groups have the highest proportion of ANC contacts (85.7%) and the WHO recommended ANC contacts. The ANC utilisation rate was higher among women having a single child (84.6%), while the WHO recommended ANC use was higher among the women having two or more children. The ANC contact rate was higher among the family with greater media exposure, but the WHO recommended ANC contact rate was relatively better among families with the least media exposure.

The religion of women showed significant association with ANC contacts as the non-Muslims women fulfilled the WHO recommended ANC contacts as well as kept the qualified doctors in contact during pregnancy. It is also evident that the mother's employment was significantly associated with ANC contacts; in contrast, the employment of spouses was significantly associated only with ANC contacts by qualified doctors.

**Table 1. Characteristics of the women, percentage of women by ANC contacts, percentage of women by the WHO recommended ANC contacts and percentage of women's ANC contacts by skilled doctors.**

| Characteristics | Number of women (%) | % of ANC contacted women | P value | % of women by ≥ 8 ANC contacts | P value | % of women's ANC contacts by qualified doctors | P value |
|---|---|---|---|---|---|---|---|
| **Total** | 4475 (100.0) | 3524 (78.4) | | 288 (8.0) | | 287 (8.1) | |
| **Geographic factors** | | | | | | | |
| *Division* | | | <0.001 | | 0.071 | | <0.001 |
| Barisal | 530 (11.8) | 75.0 | | 6.0 | | 8.0 | |
| Chittagong | 859 (19.2) | 77.0 | | 8.1 | | 9.5 | |
| Dhaka | 787 (17.6) | 84.4 | | 9.5 | | 8.8 | |
| Khulna | 530 (11.8) | 88.5 | | 10.0 | | 4.3 | |
| Rajshahi | 543 (12.1) | 78.8 | | 9.1 | | 4.6 | |
| Rangpur | 549 (12.3) | 82.5 | | 8.2 | | 9.1 | |
| Sylhet | 677 (15.1) | 64.5 | | 5.3 | | 11.9 | |
| *Place of residence* | | | <0.001 | | <0.001 | | 0.609 |
| Urban | 1440 (32.2) | 87.9 | | 12.9 | | 7.8 | |
| Rural | 3035 (67.8) | 73.9 | | 5.5 | | 8.3 | |
| **Predisposing factors** | | | | | | | |
| *Maternal Age (in years)* | | | <0.001 | | <0.001 | | 0.507 |
| <20 | 269 (6.0) | 75.7 | | 6.9 | | 8.2 | |
| 20–34 | 3269 (73.1) | 85.5 | | 11.1 | | 8.8 | |
| 35–49 | 937 (20.9) | 85.7 | | 16.7 | | 14.3 | |
| *Birth order* | | | <0.001 | | 0.003 | | 0.448 |
| 1 | 1815 (40.6) | 84.6 | | 9.1 | | 8.7 | |
| 2 | 1398 (29.9) | 79.9 | | 9.4 | | 7.4 | |
| 3 | 697 (15.6) | 75.1 | | 5.5 | | 7.6 | |
| 4 | 323 (7.2) | 67.6 | | 4.1 | | 10.5 | |
| 5–15 | 302 (6.7) | 54.0 | | 4.9 | | 6.7 | |
| *Maternal Education* | | | <0.001 | | <0.001 | | 0.014 |
| No education | 604 (13.5) | 55.8 | | 3.0 | | 6.2 | |
| Primary | 1231 (27.5) | 69.0 | | 5.9 | | 6.6 | |
| Secondary | 2121 (47.4) | 86.0 | | 7.9 | | 8.4 | |
| Higher | 519 (11.6) | 96.2 | | 16.5 | | 11.1 | |
| *Husband's education* | | | <0.001 | | <0.001 | | 0.002 |
| No education | 1024 (22.9) | 62.4 | | 4.2 | | 5.9 | |
| Primary | 1351 (30.2) | 72.5 | | 6.7 | | 7.0 | |
| Secondary | 1415 (31.6) | 87.7 | | 7.8 | | 8.5 | |
| Higher | 685 (15.3) | 94.9 | | 14.8 | | 11.3 | |
| *Religion* | | | 0.724 | | 0.016 | | 0.001 |
| Islam | 4115 (92.0) | 78.4 | | 7.8 | | 7.7 | |
| Other | 360 (8.0) | 79.2 | | 11.9 | | 13.3 | |
| **Enabling factors** | | | | | | | |
| *Media exposure* | | | <0.001 | | <0.001 | | 0.707 |
| Not at all | 1848 (41.3) | 64.6 | | 4.9 | | 7.9 | |
| Less than once a week | 393 (8.8) | 78.3 | | 7.4 | | 9.4 | |
| At least once a week | 2234 (49.9) | 64.6 | | 10.2 | | 8.1 | |
| *Wealth index* | | | <0.001 | | <0.001 | | 0.026 |
| Poorest | 937 (20.9) | 57.9 | | 3.7 | | 7.7 | |
| Poorer | 853 (19.1) | 69.7 | | 6.4 | | 7.6 | |

*(Continued)*

**Table 1.** (Continued)

| Characteristics | Number of women (%) | % of ANC contacted women | P value | % of women by ≥ 8 ANC contacts | P value | % of women's ANC contacts by qualified doctors | P value |
|---|---|---|---|---|---|---|---|
| Middle | 859 (19.2) | 80.1 | | 5.4 | | 8.1 | |
| Richer | 941 (21.0) | 88.6 | | 7.8 | | 6.3 | |
| Richest | 885 (19.8) | 96.0 | | 14.9 | | 10.6 | |
| *Mother's employment* | | | 0.023 | | 0.643 | | 0.053 |
| Working | 979 (21.9) | 78.8 | | 8.6 | | 6.4 | |
| Non-working | 3496 (78.1) | 79.2 | | 8.1 | | 8.6 | |
| *Husband's employment* | | | 0.659 | | 0.239 | | 0.005 |
| Working | 4341 (97.0) | 78.5 | | 8.3 | | 7.9 | |
| Non-Working | 134 (3.0) | 76.9 | | 4.9 | | 15.5 | |

## Multivariate analysis

From the bivariate analysis, this study considered only significant factors to present unadjusted effects of different independent variables on the frequency of ANC contacts or the WHO recommended ≥8 ANC contacts or ANC contacts by qualified doctors. Binary logistic regression models were employed to identify the adjusted effect of the explanatory variables. The adjusted effect of the explanatory variables was measured by the odds ratio along with a 95% confidence interval after controlling the effects of all other explanatory variables.

## Determinants of ANC contacts

Findings from multivariate analysis of ANC contacts were presented in Table 2. The results showed that several factors, such as administrative divisions, maternal age, birth order, mother's education, husband's education, exposure to mass media and wealth index were the most significant determinants of ANC contacts. Results revealed that respondents living in Barisal, Dhaka, Khulna, Rajshahi and Rangpur were 1.44 (95% CI: 1.09–1.91; P = 0.011), 1.94 (95% CI: 1.47–2.57; P < 0.001), 2.97 (95% CI: 2.12–4.18; P < 0.001), 1.55 (95% CI: 1.15–2.08; P = 0.004) and 2.38 (95% CI: 1.76–3.22; P < 0.001) times more likely to utilise ANC than women lived in Sylhet. Mothers, within the 20–34 years age, were 1.29 times (95% CI: 1.05–1.58; P = 0.015) more likely to use ANC than younger mothers (<20 years age group). Women with higher, secondary and primary education were 3.29 times (95% CI: 1.89–5.73; P < 0.001), 2.10 times (95% CI: 1.63–2.71; P < .001) and 1.35 times (95% CI: 1.08–1.70; P = 0.009) more likely to receive ANC services, respectively, compared to women who had no formal education. In addition, women who had husbands with secondary and higher levels of study, compared to those with non-literate husbands, were 1.49 times (95% CI: 1.17–1.90; P = 0.001) and 1.95 times (95% CI: 1.26–3.01; P = 0.003) more likely to contact ANC services, respectively.

The exposure to mass media was found to be an important determinant for receiving ANC services in the period of gestation. Women exposed to mass media were 1.69 times (95% CI: 1.37–2.09; P < 0.001) more likely to receive ANC services compared to those with no exposure to mass media. Compared to the poorest, women from richest, richer, middle and poorer families were more likely to utilise the ANC services 4.85 times (95% CI: 3.15–7.47; P < .001), 2.47 times (95% CI: 1.85–3.31; P < .001), 1.63 times (95% CI: 1.27–2.08; P < .001) and 1.29 times (95% CI: 1.05–1.59; P = 0.018), respectively. Furthermore, the number of children was negatively associated with the use of ANC services. Results suggest that women with five and more children, and two children were less likely to receive ANC services 0.66 times (95% CI: 0.49–

**Table 2. Results of binary logistic regression analysis to identify the determinants of ANC contacts (N = 4,475).**

| Factors | Estimated regression coefficient (β) | P value | Odd ratio (OR) | 95% CI of OR | |
|---|---|---|---|---|---|
| | | | | Lower | Upper |
| **Geographic factors** | | | | | |
| *Division* | | | | | |
| Barisal | 0.37 | 0.011 | 1.44 | 1.09 | 1.91 |
| Chittagong | 0.12 | 0.364 | 1.13 | 0.87 | 1.45 |
| Dhaka | 0.67 | <0.001 | 1.94 | 1.47 | 2.57 |
| Khulna | 1.09 | <0.001 | 2.97 | 2.12 | 4.18 |
| Rajshahi | 0.44 | 0.004 | 1.55 | 1.15 | 2.08 |
| Rangpur | 0.87 | <0.001 | 2.38 | 1.76 | 3.22 |
| Sylhet (ref) | 0[b] | . | . | . | . |
| *Residence* | | | | | |
| Urban | 0.17 | 0.121 | 1.18 | 0.96 | 1.46 |
| Rural (ref) | 0[b] | . | . | . | . |
| **Predisposing factors** | | | | | |
| *Maternal age (in years)* | | | | | |
| 35–49 | -0.86 | 0.462 | 0.42 | 0.04 | 4.19 |
| 20–34 | 0.25 | 0.015 | 1.29 | 1.05 | 1.58 |
| <20 (ref) | 0[b] | . | . | . | . |
| *Birth order* | | | | | |
| 5–15 | -0.42 | 0.008 | 0.66 | 0.49 | 0.90 |
| 4 | -0.21 | 0.171 | 0.81 | 0.60 | 1.09 |
| 3 | -0.11 | 0.369 | 0.90 | 0.71 | 1.14 |
| 2 | -0.22 | 0.032 | 0.80 | 0.65 | 0.98 |
| 1 (ref) | 0[b] | . | . | . | . |
| *Mother's education* | | | | | |
| Higher | 1.19 | <0.001 | 3.29 | 1.89 | 5.73 |
| Secondary | 0.74 | <0.001 | 2.10 | 1.63 | 2.71 |
| Primary | 0.30 | 0.009 | 1.35 | 1.08 | 1.70 |
| No education (ref) | 0[b] | . | . | . | . |
| *Husband's education* | | | | | |
| Higher | 0.67 | 0.003 | 1.95 | 1.26 | 3.01 |
| Secondary | 0.40 | 0.001 | 1.49 | 1.17 | 1.90 |
| Primary | 0.01 | 0.888 | 1.01 | 0.83 | 1.24 |
| No education (ref) | 0[b] | . | . | . | . |
| **Enabling factors** | | | | | |
| *Media exposure* | | | | | |
| At least one a week | 0.52 | <0.001 | 1.69 | 1.37 | 2.09 |
| Less than one a week | 0.28 | 0.054 | 1.32 | 1.00 | 1.75 |
| Not at all (ref) | 0[b] | . | . | . | . |
| *Wealth index* | | | | | |
| Richest | 1.58 | <0.001 | 4.85 | 3.15 | 7.47 |
| Richer | 0.91 | <0.001 | 2.47 | 1.85 | 3.31 |
| Middle | 0.49 | <0.001 | 1.63 | 1.27 | 2.08 |
| Poorer | 0.25 | 0.018 | 1.29 | 1.05 | 1.59 |
| Poorest (ref) | 0[b] | . | . | . | . |
| *Mother's employment* | | | | | |
| Working | -0.09 | 0.376 | 0.92 | 0.76 | 1.11 |

(*Continued*)

**Table 2.** (Continued)

| Factors | Estimated regression coefficient (β) | P value | Odd ratio (OR) | 95% CI of OR | |
|---|---|---|---|---|---|
| | | | | Lower | Upper |
| Not-working (ref) | 0[b] | . | . | . | . |
| *Husband's employment* | | | | | |
| Working | 0.27 | 0.264 | 1.30 | 0.82 | 2.08 |
| Not-working (ref) | 0[b] | . | . | . | . |

The reference category is No ANC contacts;

[b]Set to zero because it is a reference category (ref).

0.90; P = 0.008) and 0.80 times (95% CI: 0.65–0.98; P = 0.032), respectively, than those with a single child.

## Determinants of the WHO recommended ≥8 ANC contacts

Table 3 demonstrates the results of the WHO recommended ≥8 ANC contacts among women in Bangladesh. The results reveal that residence, mother's education, religion and wealth index have a significant relation with the WHO recommended ≥8 ANC contacts. Mothers from the urban areas were 1.82 times (95% CI: 1.37–2.41; P < 0.001) more likely to receive the WHO recommended ANC contacts than their rural counterparts. Women with higher levels of education were 2.86 times (95% CI: 1.30–6.29; P = 0.009) more likely to comply with the WHO recommended ANC contacts than women with no education. The odds of receiving the WHO recommended ≥8 ANC contacts among the richest women were 1.86 times (95% CI: 1.02–3.39; P = 0.043) higher than the women from poorer households. However, the likelihood of receiving the WHO recommended ANC service among Muslim women was 0.66 times (95% CI: 0.45–0.99; P = 0.043) lower than women from other religious groups.

## Determinants of ANC contacts by qualified doctors

Table 4 presents the likelihood estimation of the frequency of ANC contacts by qualified doctors. Various factors, such as administrative divisions, husband's education, religion, wealth index and husband's employment status, had a significant impact on ANC contacts by qualified doctors. Among seven administrative divisions, women who lived in Barisal, Khulna and Rajshahi divisions had 0.61 times (95% CI: 0.38–0.98; P = 0.042), 0.31 times (95% CI: 0.18–0.53; P <0.001) and 0.36 times (95% CI: 0.21–0.61; P <0.001) lower odds of contacting for ANC services by qualified doctors, respectively, compared to women lived in Sylhet division. Also, women who had higher educated husbands were 1.88 times (95% CI: 1.10–3.21; P = 0.020) more likely to receive ANC services from qualified doctors than those with the non-literate spouses. The odds of receiving ANC contacts by qualified doctors among Muslim women was 0.55 times (95% CI: 0.38–0.80; P = 0.002) lower than the women from other faith groups. Among the five categories of wealth index, only richer women had a negative association with the utilisation of ANC contacts by qualified doctors. Women, who belonged to the richer families, had 0.57 times (95% CI: 0.36–0.91; P = 0.019) less likely ANC contacts by qualified doctors compared to women from the poorest families. Husbands' working status also had a negative but significant impact on the frequency of ANC contacts by qualified doctors. Women, who had working husbands, were 0.51 times (95% CI: 0.29–0.90; P = 0.020) less likely

**Table 3. Results of binary logistic regression to determine the factors influencing the WHO recommended ≥8 ANC contacts.**

| Factors | Estimated regression coefficient (β) | P value | Odd ratio (OR) | 95% CI of OR | |
|---|---|---|---|---|---|
| | | | | Lower | Upper |
| **Geographic factor** | | | | | |
| *Residence* | | | | | |
| Urban | 0.60 | <0.001 | 1.82 | 1.37 | 2.41 |
| Rural (ref) | 0[b] | . | . | . | . |
| **Predisposing factors** | | | | | |
| *Maternal age (in years)* | | | | | |
| 35–49 | 0.26 | 0.818 | 1.30 | 0.14 | 11.86 |
| 20–34 | 0.08 | 0.581 | 1.08 | 0.81 | 1.44 |
| <20 (ref) | 0[b] | . | . | . | . |
| *Birth order* | | | | | |
| 5–15 | -0.02 | 0.967 | 0.98 | 0.46 | 2.12 |
| 4 | -0.34 | 0.346 | 0.71 | 0.35 | 1.45 |
| 3 | -0.25 | 0.250 | 0.78 | 0.51 | 1.19 |
| 2 | 0.10 | 0.475 | 1.11 | 0.84 | 1.46 |
| 1 (ref) | 0[b] | . | . | . | . |
| *Mother's education* | | | | | |
| Higher | 1.05 | 0.009 | 2.86 | 1.30 | 6.29 |
| Secondary | 0.71 | 0.051 | 2.03 | 1.00 | 4.13 |
| Primary | 0.58 | 0.113 | 1.78 | 0.87 | 3.64 |
| No education (ref) | 0[b] | . | . | . | . |
| *Husband's education* | | | | | |
| Higher | 0.36 | 0.211 | 1.43 | 0.82 | 2.51 |
| Secondary | 0.09 | 0.730 | 1.09 | 0.67 | 1.79 |
| Primary | 0.20 | 0.419 | 1.22 | 0.75 | 1.97 |
| No education (ref) | 0[b] | . | . | . | . |
| *Religion* | | | | | |
| Islam | -0.41 | 0.043 | 0.66 | 0.45 | 0.99 |
| Others (ref) | 0[b] | | | | |
| **Enabling factors** | | | | | |
| *Media exposure* | | | | | |
| At least one a week | 0.14 | 0.449 | 1.15 | 0.80 | 1.67 |
| Less than one a week | 0.11 | 0.673 | 1.12 | 0.66 | 1.90 |
| Not at all (ref) | 0[b] | . | . | . | . |
| *Wealth index* | | | | | |
| Richest | 0.62 | 0.043 | 1.86 | 1.02 | 3.39 |
| Richer | 0.27 | 0.369 | 1.30 | 0.73 | 2.33 |
| Middle | 0.08 | 0.793 | 1.08 | 0.60 | 1.96 |
| Poorer | 0.44 | 0.130 | 1.55 | 0.88 | 2.73 |
| Poorest (ref) | 0[b] | . | . | . | . |

The reference category is <8 ANC contacts;

[b]Set to zero because it is a reference category (ref).

**Table 4. Results of binary logistic regression analysis to identify the factors associated with the ANC contacts by qualified doctors.**

| Factors | Estimated regression coefficient (β) | P value | Odd ratio (OR) | 95% CI of OR | |
|---|---|---|---|---|---|
| | | | | Lower | Lower |
| **Geographic factor** | | | | | |
| *Division* | | | | | |
| Barisal | -0.50 | 0.042 | 0.61 | 0.38 | 0.98 |
| Chittagong | -0.32 | 0.119 | 0.73 | 0.49 | 1.09 |
| Dhaka | -0.34 | 0.103 | 0.71 | 0.47 | 1.07 |
| Khulna | -1.19 | <0.001 | 0.31 | 0.18 | 0.53 |
| Rajshahi | -1.03 | <0.001 | 0.36 | 0.21 | 0.61 |
| Rangpur | -0.37 | 0.102 | 0.69 | 0.44 | 1.08 |
| Sylhet (ref) | 0[b] | . | . | . | . |
| **Predisposing factors** | | | | | |
| *Mother's education* | | | | | |
| Higher | 0.41 | 0.211 | 1.51 | 0.79 | 2.86 |
| Secondary | 0.26 | 0.330 | 1.30 | 0.77 | 2.21 |
| Primary | 0.05 | 0.868 | 1.05 | 0.61 | 1.80 |
| No education (ref) | 0[b] | . | . | . | . |
| *Husband's education* | | | | | |
| Higher | 0.63 | 0.020 | 1.88 | 1.10 | 3.21 |
| Secondary | 0.43 | 0.060 | 1.54 | 0.98 | 2.43 |
| Primary | 0.21 | 0.340 | 1.24 | 0.80 | 1.92 |
| No education (ref) | 0[b] | . | . | . | . |
| *Religion* | | | | | |
| Muslim | -0.59 | 0.002 | 0.55 | 0.38 | 0.80 |
| Others (ref) | 0[b] | . | . | . | . |
| **Enabling factors** | | | | | |
| *Wealth index* | | | | | |
| Richest | -0.18 | 0.450 | 0.84 | 0.52 | 1.33 |
| Richer | -0.56 | 0.019 | 0.57 | 0.36 | 0.91 |
| Middle | -0.15 | 0.507 | 0.86 | 0.55 | 1.35 |
| Poorer | -0.13 | 0.567 | 0.88 | 0.56 | 1.38 |
| Poorest (ref) | 0[b] | . | . | . | . |
| *Mother's employment* | | | | | |
| Working | -0.22 | 0.185 | 0.80 | 0.58 | 1.11 |
| Not-working (ref) | 0[b] | . | . | . | . |
| *Husband's employment* | | | | | |
| Working | -0.67 | 0.020 | 0.51 | 0.29 | 0.90 |
| Not-working (ref) | 0[b] | . | . | . | . |

The reference category is No ANC contacts by qualified doctors;

[b]Set to zero because it is a reference category (ref).

to utilise ANC contacts by qualified doctors compared to women whose husbands were unemployed.

## Discussion

The primary purpose of this paper was to understand the geographical, predisposing and enabling factors that were associated with the ANC contacts, the WHO recommended ≥8

ANC contacts and ANC contacts by qualified doctors in Bangladesh. The study reveals that 78.4%, 8.0% and 8.1% of women in Bangladesh were receiving ANC services, the WHO recommended ≥8 ANC contacts and ANC contacts by qualified doctors, respectively. However, the rate of the WHO recommended ≥8 ANC service utilisation and ANC contacts by qualified doctors were very low compared to the ANC contacts. Findings suggest that a wide range of factors, such as the administrative divisions, maternal age, birth order, mother's education, husband's education, media exposure and wealth index were significantly associated with the ANC contacts. Residence, mother's education, religion and wealth index were also significantly associated with the WHO recommended ≥8 ANC contacts. Likewise, the administrative division, husband's education, wealth index and husband's employment were associated with ANC contacts by qualified doctors in Bangladesh. Considering the behavioural model, the association between outcome variables with independent variables were analysed in the following sections.

## Geographic factors

Findings suggest that women in Bangladesh, based on their spatial distribution, are experiencing variations in access to ANC services. Women in plain lands are more likely to access ANC services, especially contact with qualified doctors than women from mountainous regions of Bangladesh. Urban women also enjoy better access to ANC facilities compared to their counterparts from the countryside. The regional and residential variations in ANC contact, however, are not exclusive in Bangladesh as studies from other parts of the world also present similar findings. Studies in Africa suggest regional and residential, urban and rural, variations in accessing ANC [30–34]. Likewise, studies from neighbouring countries of Bangladesh confirm similar results [19, 22, 35–38]. The possible inhibiting factors, for rural and highland women, in particular, could be lack of necessary medicines, service centre as well as trained staff, long waiting time, least access to information, absence of transportation and inability to pay for the 'desired' treatment [15, 35, 37]. In urban areas as well as plain lands, on the contrary, having greater access to healthcare service facilities of both public and private- together with better social amenities and exposure to mass media for information are contributing to the better maternal wellbeing [17, 19, 35, 38–39].

## Predisposing factors

Among predisposing factors, maternal age, birth order, mother's education, husband's education and religion have a relation with the utilisation of ANC services. It is apparent that unlike teenage mothers, older women often contacted ANC services in Bangladesh, and the findings are aligned with previous studies. Some studies suggest that teenage mothers are less likely to seek ANC service [40], mainly due to the fear of social stigma [41]. In contrast, other studies indicate that younger mothers are more aware of health issues; thereby, they maintain constant contact for ANC services [23, 40–42]. Like age, women with higher education tend to seek more ANC services than women with no or least education, as found in a previous study conducted in developing countries of Asia and Africa [41]. The educational status of women was also found to be a significant predictor for the WHO recommended ≥8 ANC contacts as highly educated women complied with WHO, and the result is in agreement with studies carried out in Ethiopia [43] as well as in Bangladesh [17, 44]. Studies suggest that educated women, unlike their least educated equals, are more self-aware and capable of making decisions with confidence [15, 17, 21]. Although women's education did not significantly influence their choice of qualified doctors, the education of their spouse plays a critical role as results suggest that highly educated husbands allowed their wives to get in contact with qualified

doctors for ANC services. Studies in South Asian countries documented a positive linkage between husband's education and women's choice of qualified doctors for ANC services [17, 45, 46]. In a patriarchal society, like Bangladesh, men often have the privilege to make decisions for their wives, and in this case, the stereotypic attitude may have some effects.

In our study, we found that Muslim women were less likely to comply with the WHO recommended ANC contact than women from other religious groups, and the result has paralleled with earlier studies conducted in Bangladesh [10, 47] and beyond [48]. Muslim women's reluctance to seek ANC service could be attributed to the conservative patriarchal attitude of Muslim societies around the world. This behaviour may be the reflections of Muslim communities where religious and relevant sociocultural convictions and directions have an enormous effect [48] as women are, in general, secluded from the outside world, and sometimes they are forced to remain within the four walls of the household, especially during gestation.

### Enabling factors

Media exposure, wealth index, employment of mother and her husband, among enabling factors, played a pivotal role in seeking ANC services. Women, having exposure to mass media, contacted more often for ANC services than women with no or least acquaintance with media. A study in Ethiopia observed that higher media exposure, especially to radio and television, increases the number of ANC contacts among women [49]. Like media exposure, this study found that the economic conditions of women, measured by 'wealth index,' was a crucial predictor to comply with the WHO recommended ANC contacts, and such result was evident in previous studies [17, 47, 50]. In Bangladesh, the richest women were more likely to seek ANC compared to others. Previous studies also found a positive association between the economic status of women and ANC contact, because only the richest can pay for the health services, whereas ANC services remain underutilised by the poorest for their inability to adjust the cost of health emergencies with other family needs [30, 51–52].

Apart from wealth, employment of mothers and their husbands was assessed, and the findings indicate that only the latter had a significant relation with ANC contacts by qualified doctors. The result of this study, drawing a positive relationship between the husband's employment and ANC contacts, was coherent with earlier studies [46, 51–52]. The previous studies were suggesting that spouses involved in white-collar jobs or highly paid works often ensure more ANC contacts to qualified doctors for their wives than husbands working in blue-collar jobs.

### Strength and limitations

Several issues are determining the strengths and limitations of the current study. It is based on a nationally representative sample, covering a large sample size regarding access to ANC at regional and residential levels. The data were collected by administering globally standardized and validated research tools to conduct the interviews for quantitative analysis. This study, however, did not address the variations in smaller spatial units (sub-district or district), which may limit the interpretation of the findings at local levels. This study used the individual and socioeconomic factors, in particular, to determine the accessibility to ANC without addressing the healthcare service facilities available in the study areas through in-depth study. The cross-sectional nature of the sample design, together with recall errors as well as a tendency to provide socially desirable information by the respondents, could produce bias in the data.

## Conclusions

Despite some limitations, we can conclude that the geographical, predisposing and enabling factors are associating with the lower ANC contacts, the WHO recommended $\geq 8$ ANC contacts and ANC contacts by qualified doctors of women in Bangladesh. In order to improve women's access to antenatal care services, it is necessary to improve the literacy of women, mitigate the cost of services, increase the number of facility-based care centres and improve rural transport. The current practice of the ANC programme in Bangladesh follows the earlier WHO's guidelines of at least 4 ANC visits that proved to be a challenge for mothers in Bangladesh. In addition, the tendency to contact nonqualified healthcare providers for ANC may increase the chance of health risk for both mother and child. Therefore, the updated WHO guidelines focusing on at least eight ANC contacts and adequate ANC contacts by qualified doctors should be followed to ensure the positive pregnancy of women.

## Acknowledgments

The authors are grateful to the DHS programme for allowing us to use the BDHS data for this study, and thankful to the editor and the anonymous reviewers for their constructive suggestions and guidelines.

## Author Contributions

**Conceptualization:** Sanjoy Kumar Chanda.

**Data curation:** Sanjoy Kumar Chanda.

**Formal analysis:** Benojir Ahammed, Md. Hasan Howlader, Md Ashikuzzaman, Taufiq-E-Ahmed Shovo, Md. Tanvir Hossain.

**Methodology:** Sanjoy Kumar Chanda.

**Software:** Sanjoy Kumar Chanda, Benojir Ahammed.

**Writing – original draft:** Sanjoy Kumar Chanda, Benojir Ahammed, Md. Hasan Howlader, Md Ashikuzzaman, Taufiq-E-Ahmed Shovo, Md. Tanvir Hossain.

**Writing – review & editing:** Sanjoy Kumar Chanda, Benojir Ahammed, Md. Hasan Howlader, Md. Tanvir Hossain.

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
