## [Decision Letter · Decision Letter 0]

17 Mar 2020

PONE-D-20-02299

Factors associating antenatal care of women: A cross-sectional analysis of Bangladesh demographic and health survey 2014 data

PLOS ONE

Dear Mr. Sanjoy,

Thank you for submitting your manuscript to PLOS ONE. After careful consideration, we feel that it has merit but does not fully meet PLOS ONE’s publication criteria as it currently stands. Therefore, we invite you to submit a revised version of the manuscript that addresses the points raised during the review process.

We would appreciate receiving your revised manuscript by 6th April 2020. To enhance the reproducibility of your results, we recommend that if applicable you deposit your laboratory protocols in protocols.io, where a protocol can be assigned its own identifier (DOI) such that it can be cited independently in the future. For instructions see: http://journals.plos.org/plosone/s/submission-guidelines#loc-laboratory-protocols

We look forward to receiving your revised manuscript.

Kind regards,

Russell Kabir, PhD

Academic Editor

PLOS ONE

Journal Requirements:

2. Please refrain from stating p values as 0.00, either report the exact value or employ the format p<0.001.

3. We note you have included a table to which you do not refer in the text of your manuscript. Please ensure that you refer to Table 2, 3, 4 in your text; if accepted, production will need this reference to link the reader to the Table.

Reviewers' comments:

Reviewer's Responses to Questions

**Comments to the Author**

1. Is the manuscript technically sound, and do the data support the conclusions?

Reviewer #1: Yes

Reviewer #2: Yes

2. Has the statistical analysis been performed appropriately and rigorously? 

Reviewer #1: Yes

Reviewer #2: Yes

3. Have the authors made all data underlying the findings in their manuscript fully available?

Reviewer #1: Yes

Reviewer #2: Yes

4. Is the manuscript presented in an intelligible fashion and written in standard English?

Reviewer #1: Yes

Reviewer #2: Yes

5. Review Comments to the Author

Reviewer #1: The manuscript is a new study for Bangladesh where the ANC services are not yet up to desired level. To fulfil the SDGs goal 2030, this service must be improved, and the data presented in this study would be helpful to achieving the SDGs goal. However, there are some limitations of the study since not all administrative areas are not included in this study, and the population size is too small. The data are presented in tabular forms only which is very difficult to find out the correlation between the associated factors, so I suggest to have some pictorial presentation of the findings. The authors did not consider other associated factors such as mode of delivery, abortion, and other peri-parturient diseases etc. The teenager mothers are more involved as well interested in mass media, however, ANC service in this group of women are less than elder groups which is a contradictory issue, and not discussed well. Check the reference number 30, and correct the journal name. Others are okay.

Reviewer #2: Specific comments are as below:

1. Abstract: I would suggest to consider changing word ‘incidence’ when it was used to ‘78.4% women had the incidence of ANC contacts’ and many other occasions as it was inappropriately used in this cross sectional survey based article.

2. Last sentence in Abstract is overly complex, split this in smaller sentences to make issues more explicit.

3. 1st sentence in 2nd paragraph started with ‘NC has been defined … check what it is NC stands for…

4. 4th paragraph in Introduction ‘ Only a handful ** were conducted to find out the association between determinants and contents of ANC contacts in Bangladesh [11, 19]. All previous studies have used data from demographic and health survey: so what????.’ Pls check and restate.

5. Last sentence in Introduction is unnecessary here..

6. Pls restate the tables 2-4 heading as started as ‘Association …….

7. All the tables are placed continuously in one setting, mix them up along with the relevant texts.

8. Pls consistently use 2 digit after decimal all over..

9. In several occasions sub-heading started with ERROR! Pls check..

10. Have it read by a senior authors

6. PLOS authors have the option to publish the peer review history of their article (what does this mean?). If published, this will include your full peer review and any attached files.

Reviewer #1: No

Reviewer #2: Yes: Nazmul Alam

---

## [Author Response · Author response to Decision Letter 0]

2 Apr 2020

Reviewer#1: We have incorporated all of your suggestions into our revision. We thank you so much as comments were helpful. 

Reviewer#2: We have incorporated all of your suggestions into our revision. We thank you so much as comments were helpful.

---

## [Editor Report · Decision Letter 1]

13 Apr 2020

Factors associating different antenatal care contacts of women: A cross-sectional analysis of Bangladesh demographic and health survey 2014 data

PONE-D-20-02299R1

Dear Dr. Chanda,

We are pleased to inform you that your manuscript has been judged scientifically suitable for publication and will be formally accepted for publication once it complies with all outstanding technical requirements.

With kind regards,

Russell Kabir, PhD

Academic Editor

PLOS ONE
---

## [Editor Report · Acceptance letter]

17 Apr 2020

PONE-D-20-02299R1 

Factors associating different antenatal care contacts of women: A cross-sectional analysis of Bangladesh demographic and health survey 2014 data 

Dear Dr. Chanda:

I am pleased to inform you that your manuscript has been deemed suitable for publication in PLOS ONE. Congratulations! Your manuscript is now with our production department. 

With kind regards,

on behalf of

Dr. Russell Kabir 

Academic Editor

PLOS ONE